# Visual Transformer with Differentiable Channel Selection: An Information Bottleneck Inspired Approach

## Abstract

Self-attention and transformers have been widely used in deep learning. Recent efforts have been devoted to incorporating transformer blocks into different types of neural architectures, including those with convolutions, leading to various visual transformers for computer vision tasks. In this paper, we propose a novel and compact transformer block, Transformer with Differentiable Channel Selection, or DCS-Transformer. DCS-Transformer features channel selection in the computation of the attention weights and the input/output features of the MLP in the transformer block. Our DCS-Transformer is compatible with many popular and compact transformer networks, such as MobileViT and EfficientViT, and it reduces the FLOPs of the visual transformers while maintaining or even improving the prediction accuracy. In the experiments, we replace all the transformer blocks in MobileViT and EfficientViT with DCS-Transformer blocks, leading to DCS-Transformer networks with different backbones. The DCS-Transformer is motivated by reduction of Information Bottleneck, and a novel variational upper bound for the IB loss which can be optimized by SGD is derived and incorporated into the training loss of the network with DCS-Transformer. Extensive results on image classification and object detection evidence that DCS-Transformer renders compact and efficient visual transformers with comparable or much better prediction accuracy than the original visual transformers. The code of DCS-Transformer is available at https://anonymous.4open.science/r/IB-DCS-ViT-273C/.

## 1 Introduction

Building upon the success of Transformer in natural language processing (Vaswani et al., 2017), visual transformers have demonstrated remarkable performance compared to state-of-the-art Convolution Neural Networks (CNNs) across a wide range of computer vision tasks, including image classification (Yuan et al., 2021; Dosovitskiy et al., 2021a), object detection (Liu et al., 2021; Zhu et al., 2021), image restoration (Liang et al., 2021), and semantic segmentation (Cai et al., 2023). However, the achievements of visual transformers are accompanied by heavy computational costs (Dosovitskiy et al., 2021a; Touvron et al., 2021), making their deployment impractical under resource-limited scenarios. The aforementioned limitations have spurred recent research endeavors aimed at developing efficient visual transformers. Several approaches have been explored, including incorporation of convolution layers into visual transformers and Neural Architecture Search (NAS), to design efficient transformer architectures, and pruning redundant weights in the transformer networks. In this paper, we study the problem of compressing visual transformers by channel selection.

Channel selection is an effective method for pruning large-weight convolutional neural networks (CNNs) (Han et al., 2015; Zhou et al., 2020; Sun et al., 2021). In addition to reducing the computation cost of network architecture, adaptively selecting informative channels is also shown to be beneficial for representation learning (He et al., 2021; Han et al., 2021). Models can extract and retain essential visual information while discarding noise or irrelevant features by focusing on the most relevant channels selected for different input images. Recent works in selecting informative channels for visual transformers (Zheng et al., 2022; Chen et al., 2021b; Yu et al., 2022a) have already proven to be effective in compressing the size of the networks. However, existing works usually select fixed channels in the linear layers of transformer blocks (Yu et al., 2022b; Fang et al., 2023). In addition, they neglect the importance of selecting informative channels for calculating the affinity between their visual tokens, which is the pivotal component in aggregating global information.

**Motivation.** A typical transformer block can be written as Output $= \text{MLP}\left(\sigma(QK^\top) \times V\right)$ where $Q, K, V \in \mathbb{R}^{N \times D}$ denote the query, key, and value respectively with $N$ being the number of tokens and $D$ being the input channel number. $\sigma(\cdot)$ is an operator, such as Softmax, which generates the attention weights or affinity between the tokens. We refer to $W = \sigma(QK^\top) \in \mathbb{R}^{N \times N}$ as the affinity matrix between the tokens. MLP (multi-layer perceptron network) generates the output features of the transformer block. There are $D$ channels in the input and output features of the MLP, and $D$ is also the channel of the attention outputs. Due to the fact that the MLP accounts for a considerable amount of FLOPs in a transformer block, the size and FLOPs of a transformer block can be significantly reduced by reducing the channels of the attention outputs from $D$ to a much smaller $\tilde{D}$. **Our goal is to prune the attention output channels while maintaining and even improving the prediction accuracy of the original transformer.** However, directly reducing the channels attention outputs, even by carefully designed methods, would adversely affect the performance of the model. In this paper, we propose to maintain or even improve the prediction accuracy of a visual transformer with pruned attention outputs channels by computing a more informative affinity matrix $W$ through selecting informative channels in the query $Q$ and the key $V$. That is, only selected columns of $Q$, which correspond to the same selected rows of $K^\top$, are used to compute the affinity matrix $W = \sigma(QK^\top)$, which is refered to as channel selection for attention weights and illustrated in Figure 1a. We note that the attention outputs, which are also the input features to the MLP, is $W \times V$, and every input feature to the MLP is an aggregation of the rows of the value $V$ using the attention weights in $W$. As a result, pruning the channels of $W \times V$ amounts to pruning the channels of $V$ in the weighted aggregation. If the affinity $W$ is more informative, it is expected that a smaller number of features (rows) in $V$ contribute to such weighted aggregation, and the adverse effect of channel selection on the prediction accuracy of the transformer network is limited. Importantly, with a very informative affinity $W$, every input feature of the MLP is obtained by aggregation of the most relevant features (rows) in $V$, which can even boost the performance of visual transformers after channel selection or pruning of the attention outputs.

The idea of channel selection for the attention weights can be viewed from the perspective of Information Bottleneck (IB). Let $X$ be the input training features, $\tilde{X}$ be the learned features by the network, and $Y$ be the ground truth training labels for a classification task. The principle of IB is maximizing the mutual information between $\tilde{X}$ and $Y$ while minimizing the mutual information between $\tilde{X}$ and $X$. That is, IB encourages the network to learn features more correlated with the class labels while reducing their correlation with the input. Extensive empirical and theoretical works have evidenced that models respecting the IB principle enjoys compelling generalization. With channel selection for the attention weights, every feature in the attention outputs aggregates less features of the value $V$, so the attention outputs are less correlated with the training images so the IB principle is better adhered. This is reflected in Table 5 in Section C.2 of the supplementary, where a model for ablation study with channel selection for attention weights, DCS-Arch1 w/o IB Loss, enjoys less IB loss and higher top-1 accuracy than the vanilla transformer, MobileViT-S. It is noted that the model, DCS-Arch1 w/o IB Loss, only uses the regular cross-entropy loss in the retraining step, and smaller IB loss indicates that the IB principle is better respected. In order to further decrease the IB loss, we propose an Information Bottleneck (IB) inspired channel selection for the attention weights $\mathbf{W}$ where the learned attention weights can be more informative by explicitly optimizing the IB loss for visual transformers. Our model termed "DCS-MobileViT-S" in Table 5 is the visual transformer with the IB loss optimized, so that more informative attention weights are learned featuring even smaller IB loss and even higher top-1 accuracy. Figure 4 and Figure 5 in the supplementary illustrate the visualization results by Grad-CAM and the attention weights produced by the proposed DCS-Transformer with channel selection in attention weights, evidencing that the channel selection inspired by the IB principle leads to more informative attention weights for learning semantic concepts.

## 1.1 CONTRIBUTIONS

The contributions of this paper are presented as follows.

First, we present a novel and compact transformer block termed Transformer with Differentiable Channel Selection, or DCS-Transformer. Using our proposed channel selection in both the computation for attention weights and the features of the MLP, DCS-Transformer blocks automatically select channels in queries and keys to compute more informative attention weights inspired by the IB principle. DCS-Transformer blocks can be used to replace all the transformer blocks in many

popular visual transformers, rendering compact visual transformers with comparable or even better performance. The effectiveness of DCS-Transformer is evidenced by replacing all the transformer blocks with DCS-Transformer blocks in two visual transformers which are already compact, MobileViT (Mehta & Rastegari, 2022) and EfficientViT (Cai et al., 2023), for image classification and object detection tasks.

Second, our research is among the first few works which directly incorporate the IB loss, which is $I(\tilde{X}) - I(\tilde{X}, Y)$ where $I(\cdot, \cdot)$ denotes mutual information, into the existing training loss of a transformer network, so that the IB loss can be optimized in an end-to-end manner. In order to achieve this goal, we present a new theoretical result about a novel variational upper bound for the IB loss which can be optimized by standard SGD algorithms. Experimental results demonstrate that the IB loss of the visual transformer can be reduced by optimizing the composite loss formed by our variational upper bound for the IB loss and the regular cross-entropy loss, and the transformer network trained with such variational upper bound exhibits stronger generalization. Our variational upper bound for the IB loss is of independent interest beyond this work, and we expect that it can be broadly applied to other neural architectures so as to improve their performance by the IB principle.

We remark that as shown in Table 5 of the supplementary, channel selection in either the attention weights or the attention output without optimizing the IB loss can already reduce the IB loss. By explicitly optimizing the IB loss using its variational upper bound, network with DCS-Transformer enjoys smaller IB loss, higher classification accuracy and less FLOPs.

This paper is organized as follows. The related works in efficient visual transformers and compression of visual transformers are discussed in Section 2. The formulation of DCS-Transformer is detailed in Section 3. The effectiveness of DCS-Transformer is demonstrated in Section 4 for image classification and object detection tasks, by replacing all the transformer blocks of MobileViT and EfficientViT with DCS-Transformer blocks. We conclude the paper in Section 5.

## 2 RELATED WORKS

### 2.1 EFFICIENT VISUAL TRANSFORMERS

Recently, visual transformer models have emerged as a popular alternative to convolutional neural networks (CNNs) in computer vision tasks, such as image classification (Dosovitskiy et al., 2021b; Liu et al., 2021), object detection (Carion et al., 2020; Zhu et al., 2021), and image restoration (Liang et al., 2021; Wang et al., 2022a). Albeit the great performance of visual transformers, they usually suffer from high computation costs due to the quadratic complexity of point-wise attention modules as well as a large number of Multi-Layer Perception (MLP) encoding layers. To reduce the computation cost of visual transformer models, many recent works (Zhu et al., 2021; Yuan et al., 2021) have been proposed using different methodologies, mainly by employing sparse mechanisms to design efficient attention modules and optimizing the network architecture of the transformer models. In order to design compact network architecture of transformer models, another series of works (Cai et al., 2023; Mehta & Rastegari, 2022; Yuan et al., 2021) introduce convolutions into the network architecture of visual transformers. For instance, MobileViT (Mehta & Rastegari, 2022) introduces a hybrid architecture that combines lightweight MobileNet convolution blocks (MBConv) and MHSA modules. MobileViT places the convolution blocks at early stages in its architecture to extract low-level features while placing MHSA in late stages to achieve global representation learning. In addition, several works leverage Neural Architecture Search (NAS) (Chen et al., 2021a; Gong et al., 2022) to design efficient visual transformers. Other works also attempt to enhance the performance of efficient visual transformers by incorporating knowledge distillation into their training (Graham et al., 2021; Radosavovic et al., 2020; Gong et al., 2022).

### 2.2 COMPRESSING VISUAL TRANSFORMERS

Recent studies have also investigated compressing existing visual transformers to reduce their computation cost. Current compression methods for visual transformers usually fall into three categories: (1) Channel Pruning, which prune redundant heads and channels in ViT Blocks (Chen et al., 2021b; Chavan et al., 2022; Zheng et al., 2022). (2) Block Pruning, which drops redundant transformer blocks in transformer networks (Yu et al., 2022b;a). (3) Token Pruning, which only keeps informa-

tive tokens as the inputs for transformer blocks (Rao et al., 2021; Kong et al., 2022; Bolya et al., 2023; Wang et al., 2022b).

In this paper, we focus on channel selection for compression of visual transformers, and we propose to prune the channels of the MLP features or the attention outputs in all the transformer blocks in a visual transformer. In order to achieve comparable or even better prediction accuracy than the original visual transformer, channel selection is also performed in the computation of attention weights inspired by the IB principle so as to generate more informative attention weights, so that the adverse effect of channel reduction in the MLP features is compensated.

### 2.3 RELATED WORKS ABOUT INFORMATION BOTTLENECK

(Saxe et al., 2019) first discusses existing information bottleneck theories of deep learning. By building the connection of the compression phase of training in existing information bottleneck theories with the neural nonlinearity, they prove that the compression phase of training is not related to the excellent generalization performance of deep networks. (Lai et al., 2021) proposes to learn probabilistic maps in a spatial attention module that reduces the mutual information between the masked representation and the input while increasing the mutual information between the masked representation and the task label. (Zhou et al., 2022) proves that self-attention can be written as an iterative optimization step of the Information Bottleneck objective. Next, they show that self-attention can promote the robustness of neural networks through improved mid-level representations. They further propose a family of fully attentional networks (FANs) that take advantage of such merits of self-attention.

In contrast with most existing works that model the IB principle implicitly, our work directly optimizes the IB loss by adding its variational upper bound to the training loss of a neural network and optimizing the joint loss by standard SGD algorithms. In this manner, any neural network designed for classification tasks can enjoy potential improvement by the IB principle through the separable variational upper bound for the IB loss.

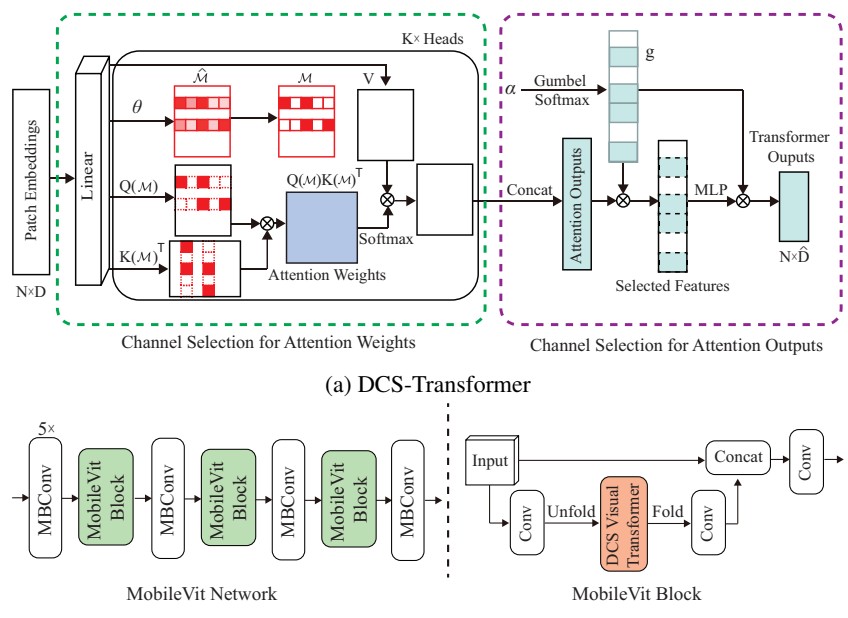

(a) DCS-Transformer

(b) Architecture of DCS-MobileViT

Figure 1: (a) A DCS-Transformer block is illustrated in (a), and (b) illustrates the architecture of MobileViT by replacing each transformer block in MobileViT with a DCS-Transformer block.

### 3 FORMULATION

In this section, we present two types of channel selection in our DCS-Transformer, which are (1) channel selection for attention weights that renders more informative attention weights or affinity be-

tween tokens; (2) channel selection for attention outputs which prunes the channels of the MLP features so as to reduce the FLOPs of the transformer block. We then present the novel variational upper bound for the IB loss, and introduce the training algorithm of the network with DCS-Transformer and the variational upper bound.

### 3.1 CHANNEL SELECTION FOR ATTENTION WEIGHTS

Given the input patch embeddings $x \in \mathbb{R}^{N \times D}$, where $N$ is the number of visual tokens and $D$ is the token dimension, visual transformer blocks first obtain the query $Q \in \mathbb{R}^{N \times D}$, key $K \in \mathbb{R}^{N \times D}$, and value $V \in \mathbb{R}^{N \times D}$ for the multi-head self-attention module with linear layers. The attention weights are then calculated by the dot-product between $Q$ and $K$. Let $x_i$ denote the $i$-th row of $x$. The $(i, j)$-th entry of the attention weight matrix $QK^\top$ is the correlation between the feature $x_i$ and $x_j$. We aim to select informative channels for the computation of the attention weights. To achieve this goal, we maintain a binary decision mask $\mathcal{M} \in \{0, 1\}^{N \times D}$, where its $(i, d)$-th element $\mathcal{M}_{id} = 1$ indicates the $d$-th channel of feature $x_i$ is selected. As a result, the attention weights or the affinity matrix $W$ is computed by $W = \sigma\left((Q \odot \mathcal{M})(K \odot \mathcal{M})^\top\right)$, where $\sigma$ is the Softmax operator on each row of the input features and $\odot$ indicates elementwise product. To optimize the discrete binary decision mask with gradient descent, we adopt simplified binary Gumbel-Softmax (Verelst & Tuytelaars, 2020) to relax $\mathcal{M} \in \{0, 1\}^{N \times D}$ into its approximation in the continuous domain $\widehat{\mathcal{M}} \in (0, 1)^{N \times D}$. The approximated soft decision mask can be computed by $\widehat{\mathcal{M}}_{id} = \sigma\left(\frac{\theta_{id} + \epsilon_{id}^{(1)} - \epsilon_{id}^{(2)}}{\tau}\right)$, where $\widehat{\mathcal{M}}_{id}$ is the $(i, d)$-th element of $\widehat{\mathcal{M}}$. $\epsilon_{id}^{(1)}$ and $\epsilon_{id}^{(2)}$ are Gumbel noise for the approximation. $\tau$ is the temperature, and $\sigma(\cdot)$ is the Sigmoid function. $\theta \in \mathbb{R}^{N \times D}$ is the sampling parameter. In this work, we obtain $\theta$ by applying a linear layer on $x$ so that the soft decision mask is dependent on the input features of the transformer block. Inspired by the straight-through estimator (Verelst & Tuytelaars, 2020; Bengio et al., 2013), we directly set $\mathcal{M} = \widehat{\mathcal{M}}$ in the backward pass. In the forward pass, the binary decision mask is computed by $\mathcal{M}_{id} = 1$ if $\widehat{\mathcal{M}}_{id} > 0.5$, and $0$ otherwise. During inference, the Gumbel noise $\epsilon_{id}^{(1)}$ and $\epsilon_{id}^{(2)}$ are set to $0$. Using the Gumbel-Softmax above, the informative channels for the attention weights computation are selected in a differentiable manner.

### 3.2 CHANNEL SELECTION FOR ATTENTION OUTPUTS

After applying the multi-head self-attention on the input patch embedding $x$, we obtain the attention outputs $z \in \mathbb{R}^{N \times D}$. Our DCS-transformer block then applies MLP layers to the attention outputs. The MLP layers in visual transformer blocks are usually computationally expensive. To improve the efficiency of the DCS-transformer, we propose to prune the channels in the attention outputs so that the computation cost of MLP layers can be reduced. Similar to the channel selection for attention weights, we maintain a decision mask $g_i \in \{0, 1\}^D$, where $g_i = 1$ indicates that the $i$-th channel is selected, and $0$ otherwise. Thus, the informative channels can be selected by multiplying $g$ by each row of the attention output. To optimize the binary decision mask with gradient descent, we replace $g$ with Gumbel Softmax weights in the continuous domain, which is computed by $\widehat{g}_i = \sigma\left(\frac{\alpha_i + \epsilon_i^{(1)} - \epsilon_i^{(2)}}{\tau}\right)$, where $\epsilon_i^{(1)}$ and $\epsilon_i^{(2)}$ are Gumbel noise. $\tau$ is the temperature. $\alpha \in \mathbb{R}^D$ is the sampling parameter. We define $\alpha$ as the architecture parameters of the DCS-Transformer block that can be optimized by gradient descent during the differentiable search process. By gradually decreasing the temperature $\tau$ in the search process, $\alpha_i$ will be optimized such that $g_i$ will approach $1$ or $0$. Note that since the MLP layers in visual transformers have the same input and output dimensions, so we multiply the decision mask $g$ with both the input and output features of the MLP layers. After the search is finished, we apply the gather operation on the attention outputs from the selected channels. The dimension of the input and output features of the MLP layers are then changed to $\tilde{D} = \sum_{i=1}^{D} g_i$. The architecture and the two types of channel selection are illustrated in Figure 1a. Our DCS-Transformer block is compatible with most visual transformers, and Figure 1b illustrates the architecture of DCS-MobileViT, which is obtained by replacing each transformer block in MobileViT with a DCS-Transformer block. In a similar manner, we can have DCS-X, where $X$ stands for a visual transformer. For example, the details about DCS-EfficientViT are introduced in Appendix A.

### 3.3 VARIATIONAL UPPER BOUND FOR THE IB LOSS

Given the training data $\{X_i, y_i\}_{i=1}^n$ where $x_i$ is the $i$-the input training feature and $y_i$ is the corresponding class label, we first specify how to compute the IB loss, $\text{IB}(\mathcal{W}) = I(\tilde{X}(\mathcal{W}), X) - I(\tilde{X}(\mathcal{W}), Y)$, where $\mathcal{W}$ is the weights of a neural network, $X$ is a random variable representing the input feature, which takes values in $\{X_i\}_{i=1}^n$, $\tilde{X}(\mathcal{W})$ is a random variable representing the learned feature which takes values in $\left\{\tilde{X}_i(\mathcal{W})\right\}_{i=1}^n$ with $\tilde{X}_i(\mathcal{W})$ being the $i$-th learned feature by the network. $Y$ is a random variable representing the class label, which takes values in $\{y_i\}_{i=1}^n$. After performing K-means clustering on $\left\{\tilde{X}_i(\mathcal{W})\right\}_{i=1}^n$ and $\{X_i\}_{i=1}^n$, we have the clusters $\{\mathcal{C}_a\}_{a=1}^A$ and $\{\mathcal{C}_b\}_{b=1}^B$ for the learned features and the input features respectively. Here we set $A = B = C$ where $C$ is the number of classes. We also abbreviate $\tilde{X}(\mathcal{W})$ as $\tilde{X}$ for simplicity of the notations. Then we define the probability that $\tilde{X}_i$ belongs to cluster $\mathcal{C}_a$ as $\Pr\left[\tilde{X} \in a\right] = \frac{1}{n}\sum_{i=1}^n \phi(\tilde{X}_i, a)$ with $\phi(\tilde{X}_i, a) = \frac{\exp\left(-\|\tilde{X}_i - \mathcal{C}_a\|_2^2\right)}{\sum_{a=1}^A \exp\left(-\|\tilde{X}_i - \mathcal{C}_a\|_2^2\right)}$. Similarly, we define the probability that $X_i$ belongs to cluster $\mathcal{C}_b$ as $\Pr[X \in b] = \frac{1}{n}\sum_{i=1}^n \phi(X_i, b)$. Moreover, we have the joint probabilities $\Pr\left[\tilde{X} \in a, X \in b\right] = \frac{1}{n}\sum_{i=1}^n \phi(\tilde{X}_i, a)\phi(X_i, b)$ and $\Pr\left[\tilde{X} \in a, Y = y\right] = \frac{1}{n}\sum_{i=1}^n \phi(\tilde{X}_i, a)\mathbb{1}_{\{y_i = y\}}$ where $\mathbb{1}_{\{\}}$ is an indicator function. As a result, we can compute the mutual information $I(\tilde{X}, X)$ and $I(\tilde{X}, Y)$ by

$$I(\tilde{X}, X) = \sum_{a=1}^A \sum_{b=1}^B \Pr\left[\tilde{X} \in a, X \in b\right] \log \frac{\Pr\left[\tilde{X} \in a, X \in b\right]}{\Pr\left[\tilde{X} \in a\right]\Pr[X \in b]},$$

$$I(\tilde{X}, Y) = \sum_{a=1}^A \sum_{y=1}^C \Pr\left[\tilde{X} \in a, Y = y\right] \log \frac{\Pr\left[\tilde{X} \in a, Y = y\right]}{\Pr\left[\tilde{X} \in a\right]\Pr[Y = y]},$$

and then compute the IB loss $\text{IB}(\mathcal{W})$. Given a variational distribution $Q(\tilde{X} \in a|Y = y)$ for $y \in \{1, \ldots C\}$ and $a \in \{1, \ldots A\}$, the following theorem gives a variational upper bound, $\text{IBB}(\mathcal{W})$, for the IB loss $\text{IB}(\mathcal{W})$.

**Theorem 3.1.**

$$\text{IB}(\mathcal{W}) \le \text{IBB}(\mathcal{W}) := \frac{1}{n}\sum_{i=1}^n \sum_{a=1}^A \sum_{b=1}^B \phi(\tilde{X}_i, a)\phi(X_i, b)\log\phi(X_i, b)$$

$$- \frac{1}{n}\sum_{i=1}^n \sum_{a=1}^A \sum_{y=1}^C \phi(\tilde{X}_i, a)\mathbb{1}_{\{y_i = y\}}\log Q(\tilde{X} \in a|Y = y) \quad (1)$$

The proof of this theorem follows by applying Lemma F.1 and Lemma F.2 in Section F of the supplementary. We remark that $\text{IBB}(\mathcal{W})$ is ready to be optimized by standard SGD algorithms because it is separable and expressed as the summation of losses on individual training points. Algorithm 1 describes the training process of a neural network with DCS-Transformer blocks where $\text{IBB}(\mathcal{W})$ is a term in the training loss. In order to compute $\text{IBB}(\mathcal{W})$ before a new epoch starts, we need to update the variational distribution $Q^{(t)}$ at the end of the previous epoch. The following functions are needed for minibatch-based training with SGD, with the subscript $j$ indicating the corresponding loss on the $j$-th batch $\mathcal{B}_j$:

$$\text{IBB}_j^{(t)}(\mathcal{W}) = \frac{1}{|\mathcal{B}_j|}\sum_{i=1}^{|\mathcal{B}_j|}\sum_{a=1}^A \sum_{b=1}^B \phi(\tilde{X}_i(\mathcal{W}), a)\phi(X_i, b)\log\phi(X_i, b) -$$

$$\frac{1}{|\mathcal{B}_j|}\sum_{i=1}^{|\mathcal{B}_j|}\sum_{a=1}^A \sum_{y=1}^C \phi(\tilde{X}_i(\mathcal{W}), a)\mathbb{1}_{\{y_i = y\}}\log Q^{(t-1)}(\tilde{X} \in a|Y = y), \quad (2)$$

$$\mathcal{L}_{\text{train},j}^{(t)}(\mathcal{W}) = \text{CE}_j^{(t)} + \eta \text{IBB}_j^{(t)}(\mathcal{W}), \ \text{CE}_j^{(t)} = \frac{1}{|\mathcal{B}_j|} \sum_{i=1}^{|\mathcal{B}_j|} H(X_i(\mathcal{W}), Y_i). \tag{3}$$

Here $\text{CE}_j^{(t)}$ is the cross-entropy loss on batch $\mathcal{B}_j$ at epoch $t$. $H(,)$ is the cross-entropy function. $\eta$ is the balance factor for the loss of information bottleneck.

---

**Algorithm 1** Training Algorithm with the Variational Upper Bound for IB by SGD

---

1: Initialize the weights of the network by $\mathcal{W} = \mathcal{W}(0)$ through random initialization
2: **for** $t \leftarrow 1$ to $t_{\text{search}}$ **do**
3:     **for** $j \leftarrow 1$ to $J$ **do**
4:         Perform gradient descent with batch $\mathcal{B}_j$ using the loss $\mathcal{L}_{\text{search},j}^{(t)}(\mathcal{W}, \alpha)$ defined Section 3.4.
5:     **end for**
6: **end for**
7: **for** $t \leftarrow 1$ to $t_{\text{train}}$ **do**
8:     **for** $j \leftarrow 1$ to $J$ **do**
9:         Update $\phi(\tilde{X}_i, a)$ for all the clusters $a \in \{1, \dots, A\}$ and $i \in \{1, \dots, n\}$.
10:         **if** $t > t_{\text{warm}}$ **then**
11:             Perform gradient descent with batch $\mathcal{B}_j$ using the loss $\mathcal{L}_{\text{train},j}^{(t)}(\mathcal{W})$ by Eq. (3).
12:         **else**
13:             Perform gradient descent with batch $\mathcal{B}_j$ using the loss $\text{CE}_j^{(t)}$ by Eq. (3).
14:         **end if**
15:     **end for**
16:     Compute $Q^{(t)}(\tilde{X} \in a | Y = y)$ by Eq. (8) in the supplementary, perform K-means clustering on $\left\{ \tilde{X}_i \right\}_{i=1}^{n}$ and update the clusters $\{\mathcal{C}_a\}_{a=1}^{A}$.
17: **end for**
18: **return** The trained weights $\mathcal{W}$ of the network

---

## 3.4 OPTIMIZATION IN THE SEARCH PROCESS

To obtain a compact visual transformer network with DCS-Transformer, we need to optimize both the accuracy of the network and the inference cost (FLOPs) of the network. Therefore, the differentiable inference cost of the network needs to be estimated and optimized during the search phase. It is worthwhile to mention that we follow the standard techniques in neural architecture search (Xie et al., 2019; Herrmann et al., 2020; Liu et al., 2019) in the searching process, including channel selection by Gumbel-Softmax and entropy minimization for architecture search. We optimize the FLOPs of the operations whose computation cost is decided by the channel selection for attention outputs in Section 3.2. For DCS-MobileViT, we estimate the FLOPs of the MLP after the channel selection on the attention outputs and the FLOPs of the convolution block following the DCS-Transformer. The estimation of the FLOPs related to a single MobileViT block is $\text{cost}_j = 2 \cdot \left( 2\tilde{D}^2 + \tilde{D} \right) + (1 + \tilde{D}) \cdot HWC$, where $j$ indexes the DCS-Transformer block. $2\tilde{D}^2 + \tilde{D}$ is the FLOPs of a layer of the MLP after the channel selection on the attention outputs, and there are two layers in the MLP. $(1 + \sum_{i=1}^{D} g_i) \cdot HWC$ is the FLOPs of the convolution block following the DCS-Transformer. $C$ is the number of filters of the convolution block. $H$ and $W$ are the height and width of the input features of the DCS-Transformer block. As a result, we can calculate the inference cost objective of the network architecture by $\text{cost} = \sum_{j=1}^{M} \text{cost}_j$, where $M$ is the number of transformer blocks. For DCS-EfficientViT, we have $\text{cost}_j = (2\tilde{D}^2 + \tilde{D}) + (1 + \tilde{D}) \cdot HWC$ which is detailed in Section A of the supplementary.

To supervise the search process, we designed a loss function incorporating cost-based regularization to enable multi-objective optimization. The overall loss function for search on each batch $\mathcal{B}_j$ at epoch $t$ is formulated by $\mathcal{L}_{\text{search},j}^{(t)}(\mathcal{W}, \alpha) = \text{CE}_j^{(t)} + \lambda \cdot \log \text{cost}(\alpha)$, $\text{CE}_j^{(t)} = \frac{1}{|\mathcal{B}_j|} \sum_{i=1}^{|\mathcal{B}_j|} H(X_i(\mathcal{W}), Y_i)$, where $\mathcal{W}$ denotes the weights in the supernet. $\alpha$ is the architecture parameters. $\lambda$ is the hyper-parameters that control the magnitude of the cost term, which is selected by cross-validation. In the search phase, the search loss is optimized to perform the two types of channel selection for all the DCS-Transformer blocks. After the search phase, we use the selected

channels for both attention weights and attention outputs in a searched network and then perform retraining on the searched network.

## 4 EXPERIMENTAL RESULTS

In this section, we first evaluate the performance of DCS-MobileViT and DCS-EfficientViT on the ImageNet-1k dataset for image classification, and show that both models render better performance than state-of-the-art networks in Section 4.1 with more compact models. In Section D, we show that networks using DCS-MobileViT and DCS-EfficientViT as the feature extraction backbones achieve better mAP with lower FLOPs than the competing baselines for object detection.

### 4.1 IMAGE CLASSIFICATION ON THE IMAGENET-1K DATASET

**Implementation details about Search/Retraining.** We use MobileViT-S (Mehta & Rastegari, 2022), MobileViT-XS (Mehta & Rastegari, 2022), and EfficientViT-B (Cai et al., 2023) as the backbones for our experiments in ImageNet classification. We replace all the transformer blocks in the backbones with DCS-Transformer blocks. In the search phase, we sample 100 classes from ImageNet (Russakovsky et al., 2015) as the training data. A cosine learning rate schedule is used in the AdamW optimizer with parameters $\beta_1$ and $\beta_2$ set to 0.9 and 0.999 respectively. The learning rate is initialized as 0.001 and then annealed to 0.0001 in 200 epochs. In each epoch, we use 70% of the training data to optimize the network weights and 30% to optimize the architecture parameters in all the DCS-Transformer blocks. We set the initial value of the temperature $\tau$ to 4.5 and decrease it by a factor of 0.95 every epoch.

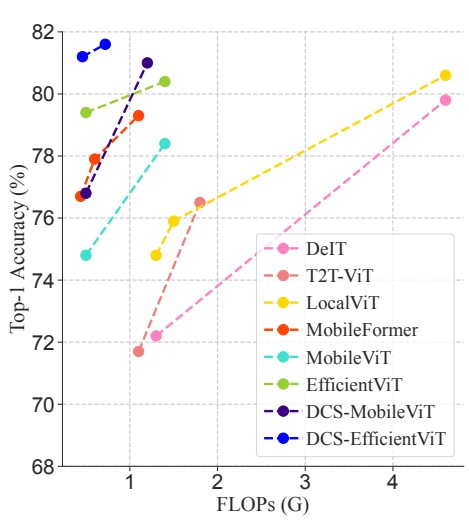

| Model | # Params | FLOPs | Top-1 |
|---|---|---|---|
| T2T | 4.3 M | 1.1 G | 71.7 |
| DeIT | 5.7 M | 1.2 G | 72.2 |
| PiT | 10.6 M | 0.7 G | 72.4 |
| CrossViT | 6.9 M | 1.6 G | 73.4 |
| MobileViT-XS | 2.3 M | 0.7 G | 74.8 |
| **DCS-MobileViT-XS (Ours)** | 2.0 M | 0.5 G | **76.8** |
| CeiT | 6.4 M | 1.2 G | 76.4 |
| DeIT | 10 M | 2.2 G | 76.6 |
| T2T | 6.9 M | 1.8 G | 76.5 |
| LocalVit | 7.7 M | 1.5 G | 76.1 |
| ConViT | 10 M | 2.0 G | 76.7 |
| PiT | 10.6 M | 1.4 G | 78.1 |
| Mobile-Former | 9.4 M | 0.5 G | 76.7 |
| EViT (Liu et al., 2023) | 12.4 M | 0.52 G | 77.1 |
| TinyViT (Wu et al., 2022) | 5.4 M | 1.3 G | 79.1 |
| DeIT | 22 M | 4.6 G | 79.8 |
| ToMe (Bolya et al., 2023) | 22 M | 2.7 G | 79.4 |
| EfficientFormer (Li et al., 2022) | 12.3 M | 1.3 G | 79.2 |
| MobileViT-S | 5.6 M | 1.4 G | 78.4 |
| VTC-LFC (Wang et al., 2022b) | 5.0M | 1.3 G | 78.0 |
| SPViT (Kong et al., 2022) | 4.9M | 1.2 G | 77.8 |
| ToMe (Bolya et al., 2023) | 5.6M | 1.2 G | 77.3 |
| **DCS-MobileViT-S (Ours)** | 4.8 M | 1.2 G | **81.0** |
| EfficientViT-B1 [r224] (Cai et al., 2023) | 9.1 M | 0.52 G | 79.4 |
| EfficientViT-B1 [r288] (Cai et al., 2023) | 9.1 M | 0.86 G | 80.4 |
| EViT (Liu et al., 2023) | 8.8 M | 0.29 G | 74.3 |
| VTC-LFC (Wang et al., 2022b) | 8.7M | 0.76 G | 79.3 |
| SPViT (Kong et al., 2022) | 8.3M | 0.71 G | 79.0 |
| ToMe (Bolya et al., 2023) | 9.1M | 0.47 G | 78.8 |
| **DCS-EfficientViT-B1 [r224] (Ours)** | 8.2 M | 0.46 G | **81.2** |
| **DCS-EfficientViT-B1 [r288] (Ours)** | 8.2 M | 0.72 G | **81.6** |

Figure 2: Top-1 accuracy vs FLOPs (G) on ImageNet-1k validation set.

Table 1: Comparisons with baseline methods on ImageNet-1k validation set.

After the search is finished, we sample the searched architecture from the supernet and perform retraining. We defer more Details about retraining, EfficientViT with DCS-Transformer, and tuning hyper-parameters by cross-validation to Section B of the supplementary. We select the values of $t_{\text{warm}}$ and $\eta$ in Algorithm 1 by cross-validation as described in Section B of the supplementary. The validation results suggest that $t_{\text{warm}} = 90$ and $\eta = 50$ work the best for all our DCS models.

**Results.** It can be observed from Table 1 that models with DCS-Transformer always enjoy less FLOPs than its original visual transformer and better accuracy. For example, DCS-EfficientViT has an accuracy improvement of almost 1% while the FLOPs is reduced from 0.54G to 0.46G compared to the original EfficientViT. In addition, DCS-MobileViT-S has an accuracy improvement of 1.5% compared to the original MobileViT-S, while its FLOPs is reduced from 1.4G to 1.2G. To study how DCS impacts the attention mechanism, we show the Grad-CAM visualization examples

in Figure 4 of the supplementary. In addition, we also visualize the attention weights computed on several examples in Figure 5 of the supplementary. To verify the effectiveness of two channel selection modules in DCS, we perform an ablation study in Section C.2 of the supplementary. More experiments on ImageNet are deferred to Section C of the supplementary. The experimental results for DCS on object detection and segmentation are presented in Section D and Section E of the supplementary.

## 4.2 STUDY IN IB LOSS AND ITS EFFECT ON GENERALIZATION

Figure 3 illustrates the test loss and the IB loss during the training for DCS-MobileVit-XS, DCS-MobileVit-S, and DCS-EfficientViT. It can be observed that the IB loss for all the models decreases starting from the 90-th epoch after the warm-up stage, and test loss of a model with IB loss optimized drops faster than that of the vanilla model.

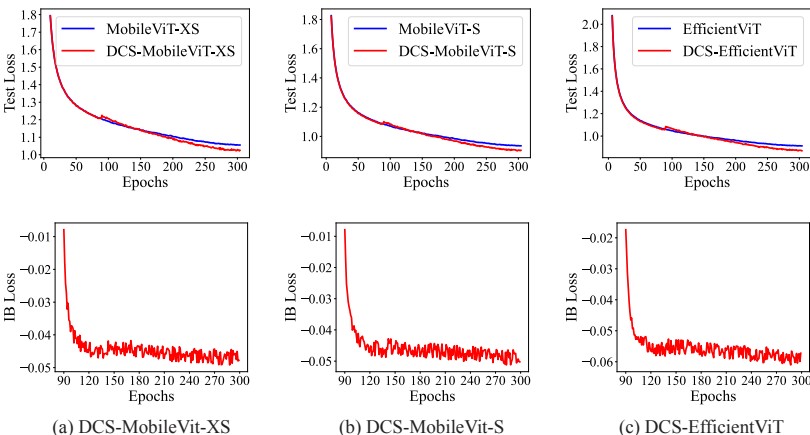

(a) DCS-MobileVit-XS   (b) DCS-MobileVit-S   (c) DCS-EfficientViT

Figure 3: Comparison: attention weights of MobileViT-S vs. DCS-MobileViT-S. The attention weights are from the first head in the last transformer block of MobileViT-S and DCS-MobileViT-S.

## 5 CONCLUSION

In this paper, we propose a novel transformer block, Transformer with Differentiable Channel Selection, or DCS-Transformer. Two types of channel selection are performed in a DCS-Transformer block, which are channel selection for attention weights and channel selection for attention outputs. The training of a network with DCS-Transformer enjoys a reduction of IB loss, rendering compact visual transformers with reduced FLOPs while enjoying comparable or even better prediction accuracy. We demonstrate the effectiveness of DCS-Transformer by replacing all the transformer blocks in MobileViT and EfficientViT with DCS-Transformer blocks, leading to DCS-MobileViT and DCS-EfficientViT, respectively. Extensive experiments on image classification and object detection demonstrate the effectiveness of DCS-Transformer, and its potential as a competitive choice when designing compact and mobile visual transformers in the future research.

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

## A  More Details about DCS-Transformer

We introduce the details about the FLOP cost of DCS-EfficientViT. For DCS-EfficientViT, we have $\texttt{cost}_j = (2\tilde{D}^2 + \tilde{D}) + (1 + \tilde{D}) \cdot HWC$. The first term, $2\tilde{D}^2 + \tilde{D}$, is the FLOPs of the linear layer at the end of the Lightweight Multi-Scale Attention block, $(1+\tilde{D}) \cdot HWC$ is the FLOPs of the first convolution layer in the MBConv block after the lightweight Multi-Scale Attention (lightweight MSA) block. $\tilde{D}$ denotes the compressed input dimension of the linear layer at the end of the lightweight MSA block. Note that $\tilde{D}$ is compressed from concatenated features of $3 \cdot D$ dimensions, where $D$ is the output dimension of the ReLU Global Attention and $3$ is the number of branches in Lightweight Multi-Scale Attention block in EfficientViT.

## B  More Details about Searching/Retraining

**More Details about Retraining and EfficientViT with DCS-Transformer.** In the retraining phase, we train the searched network on ImageNet-1K. We also use AdamW optimizer and set $\beta_1$ and $\beta_2$ to 0.9 and 0.999. The retraining phase takes 300 epochs, whereas the first 90 epochs are the warm-up stage that only optimizes the cross-entropy loss. After the warm-up stage, we optimize the overall training loss function that minimizes the cross-entropy loss and the IB loss. All the training is performed on 4 NVIDIA V100 GPUs with an effective batch size of 512 images. Following previous works (Cai et al., 2023), we adopt the widely used and standard data augmentation techniques, including random scaling, random horizontal flip, and random cropping in training. The weight decay is set to 0.01. The learning rate is increased from 0.0002 to 0.002 linearly in the first 5 epochs and then annealed to 0.0002 using a cosine learning rate schedule in the following 300 epochs. The inference is performed on the exponential moving average of model weights. Different from works such as LeViT (Graham et al., 2021) and NASVIT (Gong et al., 2022), knowledge distillation is not used in the training of DCS models. Our results are compared exclusively with baseline models trained without knowledge distillation.

For transformer blocks in EfficientViT, we apply our method of channel selection for attention weights, illustrated in the green box in Figure 1a, on the queries and keys for the ReLU Global Attention in every branch of the lightweight MSA module before applying ReLu and dot-product on them. We also apply our channel selection method for attention outputs to the features concatenated from different scale branches and prune the number of channels in the following linear layer.

**Tuning Hyper-Parameters by Cross-Validation.** To decide the best balancing factor $\lambda$ for the overall search loss, $\eta$ for the overall training loss, and the number of warm-up epochs $t_{\text{warm}}$, we perform 5-fold cross-validation on $10\%$ of the training data. The value of $\lambda$ is selected from $\{0.1, 0.15, 0.2, 0.25, 0.3, 0.35, 0.4, 0.45, 0, 5\}$. The value of $\eta$ is selected from $\{0.1, 0.5, 1, 5, 10, 50, 100\}$. The value of $t_{\text{warm}}$ is selected from $\{0.1t_{\text{train}}, 0.2t_{\text{train}}, 0.3t_{\text{train}}, 0.4t_{\text{train}}, 0.5t_{\text{train}}, 0.6t_{\text{train}}\}$, where $t_{\text{train}} = 300$ is the total number of training epochs. We select the values of $\eta$, $\lambda$, and $t_{\text{warm}}$ that lead to the smallest validation loss. It is reveled that $t_{\text{warm}} = 90$ is chosen for all the three visual transformers in our experiments.

## C  More Experiments

We provide additional results of visual transformers with DCS-Transformer blocks which are trained by knowledge distillation. Existing efficient visual transformers, such as NASViT (Gong et al., 2022) and LeViT (Graham et al., 2021), incorporate knowledge distillation to improve their performance. They employ the outputs of large models as supervision during their own training. In this section, we adopt knowledge distillation in the retraining process of our models. To ensure fair comparisons, we follow the settings in NASViT and employ EfficientNet-B5 as the teacher model for knowledge distillation. The results are shown in Table 2, where KD denotes our models trained with knowledge distillation. Furthermore, to compare with previous SOTA methods NASViT-A4 and NASViT-A5, we also evaluate DCS-EfficientViT on an input resolution of $288 \times 288$, which is denoted by $r288$ in the table. It is shown in the results that DCS-EfficientViT trained with both input resolutions outperforms NASViT in terms of higher accuracy while enjoying lower FLOPs. For example, with input resolution of $288 \times 288$, DCS-EfficientViT has a lower FLOPs (0.72G compared to 0.76G) while its accuracy is higher ($83.1\%$ compared to $81.8\%$).

| Model | # Params | FLOPs | Top-1 |
|---|---|---|---|
| EfficientViT-B1 [r224] (Cai et al., 2023) | 9.1 M | 0.54 G | 79.4 |
| EfficientViT-B1 [r288] (Cai et al., 2023) | 9.1 M | 0.86 G | 80.4 |
| EViT (Liu et al., 2023) | 8.8 M | 0.3 G | 74.3 |
| VTC-LFC (Wang et al., 2022b) | 8.7 M | 0.76G | 79.3 |
| SPViT (Kong et al., 2022) | 8.3 M | 0.71G | 79.0 |
| ToMe (Bolya et al., 2023) | 9.1 M | 0.47G | 78.8 |
| **DCS-EfficientViT-B1 [r224] (Ours)** | 8.2 M | 0.46 G | **81.2** |
| **DCS-EfficientViT-B1 [r288] (Ours)** | 8.2 M | 0.72 G | **81.6** |
| LeViT-128 (Graham et al., 2021) | 9.2 M | 0.41 G | 78.6 |
| LeViT-192 (Graham et al., 2021) | 11.0 M | 0.67 G | 80.0 |
| LeViT-256 (Graham et al., 2021) | 19.0 M | 1.12 G | 81.6 |
| NASViT-A2 (r224) (Gong et al., 2022) | 16.5 M | 0.42 G | 80.5 |
| NASViT-A3 (r256) (Gong et al., 2022) | 16.5 M | 0.53 G | 81.0 |
| NASViT-A4 (r288) (Gong et al., 2022) | 16.5 M | 0.59 G | 81.4 |
| NASViT-A5 (r288) (Gong et al., 2022) | 16.5 M | 0.76 G | 81.8 |
| **DCS-EfficientViT-B1 [r224, KD] (Ours)** | 8.2 M | 0.46 G | **82.1** |
| **DCS-EfficientViT-B1 [r288, KD] (Ours)** | 8.2 M | 0.72 G | **83.1** |

Table 2: Comprehensive Comparisons on ImageNet-1k validation set.

| Model | # Params | FLOPs | Top-1 |
|---|---|---|---|
| MobileViT-S | 5.6 | 1.6 | 78.4 |
| DCS-Arch1 (w/o IB Loss) | 5.6 | 1.6 | 79.4 |
| DCS-Arch2 (w/o IB Loss) | 4.8 | 1.2 | 78.2 |
| DCS-QK-Mask (w/o IB Loss) | 4.8 | 1.2 | 79.7 |
| DCS-MobileViT-S (w/o IB Loss) | 4.8 | 1.2 | 79.9 |
| DCS-MobileViT-S | 4.8 M | 1.2 G | **81.0** |

Table 3: Study on the mask mechanism of DCS MobileViT-S.

We also introduce another ablation model named DCS-QK-Mask where different binary decision masks are used in $Q$ and $K$ when computing the attention weights $W$, that is, $W = \sigma\left((Q \odot \mathcal{M}_1)(K \odot \mathcal{M}_2)^\top\right)$ where $\mathcal{M}_1$ and $\mathcal{M}_2$ are learned separately in the search phase. As shown in Table 3, the DCS-MobileViT still outperforms DCS-QK-Mask using MobileViT as the backbone, and this is because different decision masks introduce unnecessary complexity in the self-attention module.

Figure 6 illustrates the histogram of the entropy of the rows of the affinity matrices over randomly chosen 100 validation images of ImageNet-1k for all the four heads in the last transformer block in MobileViT-S and DCS-MobileViT-S. It can be observed that more rows of the affinity matrices for DCS-Transformer have lower entropy, indicating that the affinity matrices in the DCS-Transformer blocks are more informative.

## C.1 VISUALIZATION RESULTS

In this section, we first apply the Grad-CAM (Selvaraju et al., 2017) visualization tool to study which parts in the input images are responsible for the predictions of the baseline models and models. Figure 4 shows the heatmaps of different models generated by Grad-CAM. It can be observed that our DCS models focused more on the object for classification in the input images. In contrast, the activation maps generated by the baseline models also have nonnegligible values in the background.

To study how DCS impacts the attention mechanism, we visualize the attention weights computed for some specific queries in the transformer blocks in both MobilViT-S and DCS-MobileViT-S in Figure 5.

## C.2 ABLATION STUDY

In order to demonstrate the effects of the two types of channel selection, channel selection for attention weights and channel selection for attention outputs described in Section 3.1 and Section 3.2 respectively, we design two ablation DCS-Transformer modules, which are DCS-Arch1 and DCS-

| Block | Block 1 | Block 2 | Block 3 |
|---|---|---|---|
| N | 196 | 49 | 16 |
| D | 36 | 48 | 60 |
| r | 34.2, 34.5, 34.7, 35.2 | 42.1, 42.7, 43.0, 43.2 | 46.9, 47.7, 50.0, 51.3 |

Table 4: The rank $r = \mathrm{rank}((Q \odot \mathcal{M})(K \odot \mathcal{M})^\top)$ for all the attention heads for each DCS-MobileViT-S block.

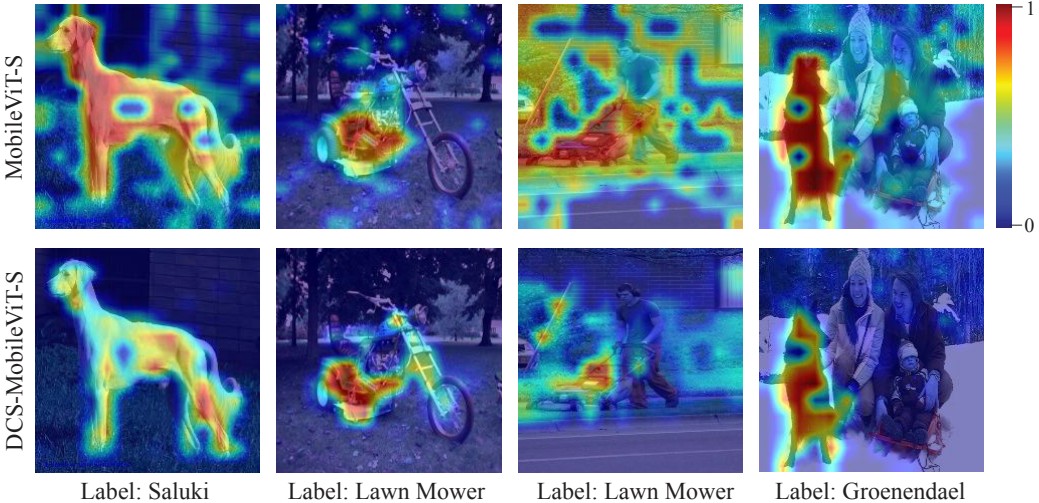

Label: Saluki  Label: Lawn Mower  Label: Lawn Mower  Label: Groenendael

Figure 4: Grad-CAM visualization results. The figures in the first row are the visualization results of MobileViT-S, and the figures in the second row are the visualization results of DCS-MobileViT-S.

Arch2. DCS-Arch1 only has channel selection for attention weights, and DCS-Arch2 only has channel selection for attention outputs. We replace all the transformer blocks in MobileViT-S with one of these two ablation DCS-Transformer modules, respectively, and report their performance on the ImageNet-1k dataset in Table 5. It can be observed that DCS-Arch2 reduces the FLOPs of the model by pruning the channels of the MLP features at a cost of loss in accuracy. In contrast, the proposed channel selection for attention weights significantly boosts the accuracy of DCS-MobileViT-S compared to that of the original MobileViT-S by 2.6% while enjoying lower FLOPs.

| Model | # Params | FLOPs | Top-1 | IB Loss |
|---|---|---|---|---|
| MobileViT-S | 5.6 | 1.6 | 78.4 | -0.00432 |
| DCS-Arch1 (w/o IB Loss) | 5.6 | 1.6 | 79.4 | -0.00459 |
| DCS-Arch2 (w/o IB Loss) | 4.8 | 1.2 | 78.2 | -0.00446 |
| DCS-MobileViT-S (w/o IB Loss) | 4.8 | 1.2 | 79.9 | -0.00475 |
| DCS-MobileViT-S | 4.8 | 1.2 | 81.0 | -0.05122 |

Table 5: Ablation Study of DCS MobileViT-S.

### C.3 COMPARISON OF ATTENTION WEIGHTS

As explained in the introduction and Section 3.1, the channel selection for attention weights is expected to generate a more informative affinity matrix $W$. It is noted that every row of the affinity matrix $\mathbf{W}$ is a normalized nonnegative vector with all elements summing to 1, which represents the affinity scores of a certain query to each key. Figure 7 illustrates the affinity matrix $W$ of the first head in the last transformer block of MobileViT-S and DCS-MobileViT-S trained on the ImageNet-1k dataset, using a particular image sampled from the training data as the input. More spiked patterns are observed for DCS-MobileViT-S, indicating that fewer keys contribute to the

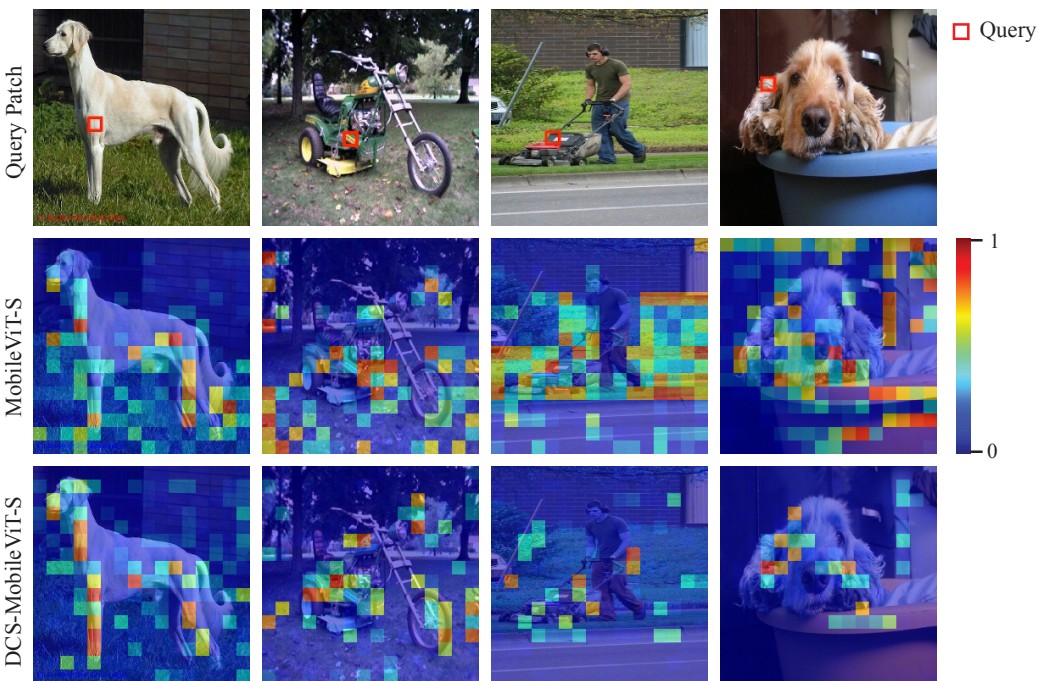

Figure 5: Visualization of attention weights. The query patch positions for all examples are marked with red bounding boxes in the first row. We visualize the attention weights of the first head in the first transformer block in MobileViT-S (second row) and DCS-MobileViT-S (third row). The original input image in the shape of $256 \times 256$ corresponds to $16 \times 16$ non-overlapped patches for self-attention, where each patch corresponds to $16 \times 16$ pixels in the input image. We visualize the attention weights between the query patch and all $16 \times 16$ patches.

feature aggregation process for computing the attention outputs. As a result, channel selection on the attention outputs has less adverse effect on the performance of the resultant model. We also adopt entropy as a quantitative measure for the information in the affinity matrix $W$. We show the average entropy of all the rows of the matrix $\mathbf{W}$ for the first head in the last transformer block of MobileViT-S and DCS-MobileViT-S in Figure 7, and a lower average entropy of $1.84$ compared to $2.50$ indicates that the affinity matrix in a DCS-Transformer block is more informative than its counterpart. We also compare such average entropy of the affinity matrix over all the validation images of ImageNet-1k for each the four heads in the last transformer block in MobileViT-S and DCS-MobileViT-S, evidencing that DCS-Transformer renders more informative affinity: Head 1, Baseline 2.27 vs. DCS 2.09; Head 2, Baseline 2.18 vs. DCS 2.05; Head 3, Baseline 2.33 vs. DCS 2.15; Head 4, Baseline 2.24 vs. DCS 2.08. More details about attention weights and the histograms about the average entropy of the affinity matrix are deferred to Section C of the supplementary.

## C.4 COMPARISON OF ATTENTION WEIGHTS

## D OBJECT DETECTION

**Implementation details.** We integrate ImageNet pre-trained DCS-MobileViT-XS, DCS-MobileViT-S, and DCS-EfficientViT with the single-shot object detection backbone SSDLite (Sandler et al., 2018). The models are evaluated on the MS-COCO dataset (Lin et al., 2014) that contains 117k training and 5k validation images. We fine-tune all pre-trained DCS-Transformer networks with the object detection framework at an input resolution of $320 \times 320$. All models are trained with AdamW for 200 epochs following the same settings in MobileViT (Mehta & Rastegari, 2022). With a cosine learning rate scheduler, the learning rate is decayed from $0.0009$ to $1.6e^{-6}$. We use smooth $\ell^1$ for the object localization objective and cross-entropy losses for classification. The performance is evaluated on the validation set using mAP@IoU of 0.50:0.05:0.95.

**Results.** We conduct a comparative analysis of our models against alternative lightweight feature backbones that are integrated with the SSDLite object detection framework. The results are shown in Table 6 of the supplementary, which show that object detection performance is greatly improved by replacing the feature backbone with models with DCS-Transformer blocks. For instance, the mAP is improved by 1% after replacing MobilViT-S with DCS-MobilViT-S while saving 0.3G FLOPs. Furthermore, with the same FLOPs SSDLite as MobilNetV3, SSDLite with DCS-EfficientViT achieves performance improvement of 7%.

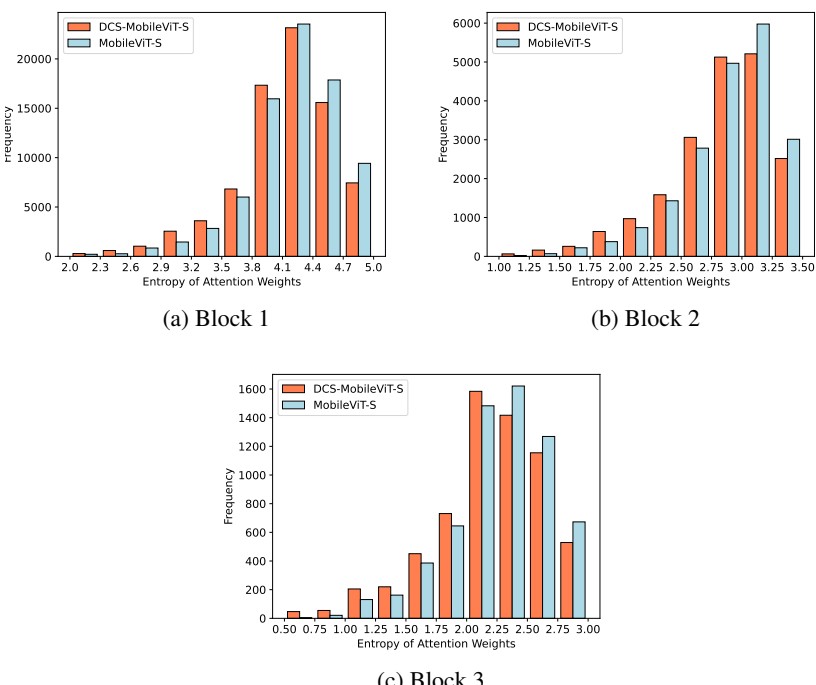

(a) Block 1        (b) Block 2

(c) Block 3

Figure 6: Comparisons on entropy for attention weights of query tokens from all attention heads for each of the three DCS-Transformer blocks in MobileVit-S and DCS-MobileVit-S. The results are accumulated from 100 randomly sampled images from the validation set of ImageNet-1k.

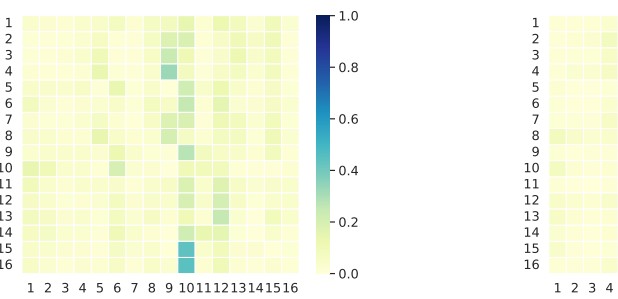

(a) Attention weights of MobileViT-S. The average entropy of attention weights for all queries is 2.50.

(b) Attention weights of DCS-MobileViT-S. The average entropy of attention weights for all queries is 1.84.

Figure 7: Comparisons between attention weights of MobileViT-S and DCS-MobileViT-S. The attention weights are from the first head in the last transformer block of MobileViT-S and DCS-MobileViT-S.

| Feature backbone | # Params. | FLOPs | mAP |
|---|---|---|---|
| MobileNetv3 | 4.9 M | 1.4 G | 22.0 |
| MobileNetv2 | 4.3 M | 1.6 G | 22.1 |
| MobileNetv1 | 5.1 M | 2.6 G | 22.2 |
| MixNet | 4.5 M | 2.2 G | 22.3 |
| MNASNet | 4.9 M | 1.7 G | 23.0 |
| YoloV5-N (640×640) | 1.9 M | 4.5 G | 28.0 |
| Vidt (Song et al., 2022) | 7.0 M | 6.7 G | 28.7 |
| MobileViT-XS | 2.7 M | 1.7 G | 24.8 |
| **DCS-MobileViT-XS(Ours)** | 2.4 M | 1.5 G | **25.8** |
| MobileViT-S | 5.7 M | 2.4 G | 27.7 |
| **DCS-MobileViT-S(Ours)** | 4.7 M | 2.1 G | **29.0** |
| EfficientViT | 9.9 M | 1.5 G | 28.4 |
| **DCS-EfficientViT(Ours)** | 9.0 M | 1.4 G | **29.3** |

Table 6: Object detection performance with SSDLite.

# E  INSTANCE SEGMENTATION

In this section, we evaluate the performance of DCS for the task of instance segmentation on the COCO (Lin et al., 2014) dataset. We adopt Mask R-CNN (He et al., 2017) with Feature Pyramid Network (FPN) as the segmentation head on top of the feature backbone of DCS-EfficientViT-B1. We include EfficientViT-B1 (Cai et al., 2023) and EViT (Liu et al., 2023) as baselines for comparisons. We train our model and the baselines on the train split of the dataset and report the performance on the validation split following (Chen et al., 2019). Both our method and the baselines are trained for 12 epochs following the settings of $1\times$ schedule in (Chen et al., 2019). We use AdamW as the optimizer in the training following (Liu et al., 2023). The initial learning rate is set to 0.001 and decays with a cosine learning rate schedule. We measure and report the mean bounding box Average Precision (mAP$^b$) and mean mask Average Precision (mAP$^b$) as well as bounding box Average Precision (AP$^b$) and mask Average Precision (AP$^b$) under IoU thresholds of 0.5 and 0.75. The results are shown in Table 7. It can be observed that DCS-EfficientViT-B1 consistently improves the performance of segmentation across various thresholds.

| Methods | mAP$^{box}$ | AP$^b_{50}$ | AP$^b_{75}$ | mAP$^m$ | AP$^m_{50}$ | AP$^m_{75}$ |
|---|---|---|---|---|---|---|
| EViT (Liu et al., 2023) | 32.8 | 54.4 | 34.5 | 31.0 | 51.2 | 32.2 |
| EfficientViT-B1 (Cai et al., 2023) | 33.5 | 55.4 | 34.8 | 31.9 | 52.3 | 32.7 |
| DCS-EfficientViT-B1 | **34.8** | **56.3** | **35.3** | **33.2** | **53.1** | **33.3** |

Table 7: Instance Segmentation Results on COCO.

# F  PROOF OF THEOREM 3.1

**Lemma F.1.**

$$I(\tilde{X}, X) \leq \frac{1}{n} \sum_{i=1}^{n} \sum_{a=1}^{A} \sum_{b=1}^{B} \phi(\tilde{X}_i, a)\phi(X_i, b) \log \phi(X_i, b) - \frac{1}{n^2} \sum_{i=1}^{n} \sum_{j=1}^{n} \sum_{b=1}^{B} \phi(X_i, b) \log \phi(X_j, b)$$

(4)

*Proof.* By the log sum inequality, we have

$$I(\tilde{X}, X)$$

$$= \sum_{a=1}^{A} \sum_{b=1}^{B} \Pr\left[\tilde{X} \in a, X \in b\right] \log \frac{\Pr\left[\tilde{X} \in a, X \in b\right]}{\Pr\left[\tilde{X} \in a\right] \Pr\left[X \in b\right]}$$

$$\leq \frac{1}{n^2} \sum_{i=1}^{n} \sum_{j=1}^{n} \sum_{a=1}^{A} \sum_{b=1}^{B} \phi(\tilde{X}_i, a)\phi(X_i, b) \left(\log\left(\phi(\tilde{X}_i, a)\phi(X_i, b)\right) - \log\left(\phi(\tilde{X}_i, a)\phi(X_j, b)\right)\right)$$

$$= \frac{1}{n^2} \sum_{i=1}^{n} \sum_{j=1}^{n} \sum_{a=1}^{A} \sum_{b=1}^{B} \phi(\tilde{X}_i, a)\phi(X_i, b) \log \phi(X_i, b)$$

$$- \frac{1}{n^2} \sum_{i=1}^{n} \sum_{j=1}^{n} \sum_{a=1}^{A} \sum_{b=1}^{B} \phi(\tilde{X}_i, a)\phi(X_i, b) \log \phi(X_j, b)$$

$$= \frac{1}{n} \sum_{i=1}^{n} \sum_{a=1}^{A} \sum_{b=1}^{B} \phi(\tilde{X}_i, a)\phi(X_i, b) \log \phi(X_i, b)$$

$$- \frac{1}{n^2} \sum_{i=1}^{n} \sum_{j=1}^{n} \sum_{a=1}^{A} \sum_{b=1}^{B} \phi(\tilde{X}_i, a)\phi(X_i, b) \log \phi(X_j, b). \tag{5}$$

$\square$

**Lemma F.2.**

$$I(\tilde{X}, Y) \geq \frac{1}{n} \sum_{a=1}^{A} \sum_{y=1}^{C} \sum_{i=1}^{n} \phi(\tilde{X}_i, a) \mathbb{1}_{\{y_i = y\}} \log Q(\tilde{X} \in a | Y = y) \tag{6}$$

*Proof.* Let $Q(\tilde{X}|Y)$ be a variational distribution. We have

$I(\tilde{X}, Y)$

$$= \sum_{a=1}^{A} \sum_{y=1}^{C} \Pr\left[\tilde{X} \in a, Y = y\right] \log \frac{\Pr\left[\tilde{X} \in a, Y = y\right]}{\Pr\left[\tilde{X} \in a\right] \Pr[Y = y]}$$

$$= \sum_{a=1}^{A} \sum_{y=1}^{C} \Pr\left[\tilde{X} \in a, Y = y\right] \log \frac{\Pr\left[\tilde{X} \in a|Y = y\right] Q(\tilde{X} \in a|Y = y)}{\Pr\left[\tilde{X} \in a\right] Q(\tilde{X} \in a|Y = y)}$$

$$\geq \sum_{a=1}^{A} \sum_{y=1}^{C} \Pr\left[\tilde{X} \in a, Y = y\right] \log \frac{\Pr\left[\tilde{X} \in a|Y = y\right]}{Q(\tilde{X} \in a|Y = y)} + \sum_{a=1}^{A} \sum_{y=1}^{C} \Pr\left[\tilde{X} \in a, Y = y\right] \log \frac{Q(\tilde{X} \in a|Y = y)}{\Pr\left[\tilde{X} \in a\right]}$$

$$= \mathrm{KL}\left(P(\tilde{X}|Y)\big\|Q(\tilde{X}|Y)\right) + \sum_{a=1}^{A} \sum_{y=1}^{C} \Pr\left[\tilde{X} \in a, Y = y\right] \log \frac{Q(\tilde{X} \in a|Y = y)}{\Pr\left[\tilde{X} \in a\right]}$$

$$\geq \sum_{a=1}^{A} \sum_{y=1}^{C} \Pr\left[\tilde{X} \in a, Y = y\right] \log \frac{Q(\tilde{X} \in a|Y = y)}{\Pr\left[\tilde{X} \in a\right]}$$

$$= \sum_{a=1}^{A} \sum_{y=1}^{C} \Pr\left[\tilde{X} \in a, Y = y\right] \log Q(\tilde{X} \in a|Y = y) + H\left(P(\tilde{X})\right)$$

$$\geq \sum_{a=1}^{A} \sum_{y=1}^{C} \Pr\left[\tilde{X} \in a, Y = y\right] \log Q(\tilde{X} \in a|Y = y)$$

$$\geq \frac{1}{n} \sum_{a=1}^{A} \sum_{y=1}^{C} \sum_{i=1}^{n} \phi(\tilde{X}_i, a) \mathbb{I}_{\{y_i = y\}} \log Q(\tilde{X} \in a|Y = y). \tag{7}$$

$\square$

## F.1 COMPUTATION OF $Q^{(t)}(\tilde{\mathbf{X}}|Y)$

## G SEC:Q-COMPUTE

The variational distribution $Q^{(t)}(\tilde{\mathbf{X}}|Y)$ can be computed by

$$Q^{(t)}(\tilde{X} \in a|Y = y) = \Pr\left[\tilde{X} \in a|Y = y\right]$$

$$= \frac{\sum_{i=1}^{n} \phi(\tilde{X}_i, a) \mathbb{I}_{\{y_i = y\}}}{\sum_{i=1}^{n} \mathbb{I}_{\{y_i = y\}}}. \tag{8}$$

