# OpenReview forum: "Visual Transformer with Differentiable Channel Selection: An Information Bottleneck Inspired Approach"
_ICLR.cc/2024/Conference — Submitted to ICLR 2024_

### Official Review · Reviewer_wQJ5 · 2023-10-31

**Soundness:** 3 good
**Presentation:** 3 good
**Contribution:** 2 fair
**Rating:** 5
**Confidence:** 4

**Summary:**

The study introduces an innovative and streamlined transformer block named DCS-Transformer, which facilitates channel selection for both attention weights and attention outputs. The inspiration for this channel selection arises from the information bottleneck (IB) principle. This principle strives to diminish the mutual information between the transformer block's input and output, all the while maintaining the mutual information between the output and the task label. This is chiefly realized through the use of a Gumbel-softmax and a channel-pruning loss.

The overall framework mirrors the structure presented in the Neural Architecture Search (NAS), encompassing both a search phase and a training phase. In the quest for the optimal weights for channel selection, the authors put forth an IB-associated loss for the search process. The proficiency of the presented approach is corroborated by experimental findings on ImageNet-1k and COCO.

**Strengths:**

1. The paper is well written.

2. The introduction of IB loss into Vision Transformers sounds novel.

**Weaknesses:**

1. The rationale for utilizing an Information Bottleneck loss appears to be somewhat rigid and unclear to me.
- The authors explain that the reason for employing this loss is that


> IB prompts the network to learn features that are more strongly correlated with class labels while decreasing their correlation with the input.


However, it remains unclear to me why a traditional Softmax cross-entropy loss wouldn't be adequate to address this issue effectively.

2. The primary contribution of this paper doesn't seem to be particularly effective.

As per Table 6 in the appendix, the implementation of the proposed IB loss results in only a 0.3% improvement when used on the same backbone. Such a minor improvement could also be attained simply by using a more favourable random seed, which might be too trivial to serve as the main contribution of a ICLR paper.

3. Some of the ablation studies on hyper-parameter search are missing.

I'm intrigued by the roles of the hyper-parameters $\eta$ and $\lambda$ in the suggested approach. It appears that if $\eta$ is not set to a small value, the outcomes could be inferior to the baseline. Upon examining the code, I noticed that $\eta$ is defaulted to 0.1, which contrasts with the paper's assertion that $\eta$ is set to 50 for ImageNet. This discrepancy could potentially be confusing for many readers.

4. The discussion on related works is not comprehensive enough.

This paper introduces some techniques, e.g. channel selection with Gumbel SoftMax, and entropy minimization for architecture search, that were first applied in the field of Neural Architecture Search (NAS) and network pruning. However, the section of related work does not include a subsection in this direction, which is inappropriate from my point of view. Some seminal works like [1, 2, 3] should be included and carefully discussed.

### Reference

[1] Xie S, Zheng H, Liu C, et al. SNAS: stochastic neural architecture search.ICLR 2019.

[2] Herrmann C, Bowen R S, Zabih R. Channel selection using gumbel softmax. ECCV 2020.

[3] Liu H, Simonyan K, Yang Y. Darts: Differentiable architecture search. ICLR 2019

**Questions:**

see weaknesses

---

> ### Author Response · Authors · 2023-11-23
> **Response to Reviewer wQJ5 Part 1**
>
> We appreciate the review and the suggestions in this review. The raised issues are addressed below.
>
> (1) **More detailed motivation for the IB Loss and Regular Cross-Entropy Cannot Decrease the IB Loss Enough**
>
>
> More detailed motivation for the incorporate of information bottleneck loss is presented in Section 1 of the revised paper.
> It is attached below for your convenience.
>
> **Motivation**
>
> A typical transformer block can be written as
> $\texttt{Output} = \texttt{MLP} \left(\sigma(QK^{\top})\times V \right)$
> where $Q,K,V \in \mathbb{R}^{N \times D}$ denote the query, key, and value respectively
> with $N$ being the number of tokens and $D$ being the input channel number.
> $\sigma(\cdot)$ is an operator, such as Softmax, which generates the attention weights or affinity between the tokens.
> We refer to $W  = \sigma(QK^{\top}) \in \mathbb{R}^{N \times N}$ as the affinity matrix between the tokens.
> $\textup{MLP}$ (multi-layer perceptron network) generates the output features of the transformer block.
> There are $D$ channels in the input and output features of the MLP, and $D$ is also the channel of the attention outputs.
> Due to the fact that the MLP accounts for a considerable amount of FLOPs in a transformer block, the size and FLOPs of a transformer
> block can be significantly reduced by reducing the channels of the attention outputs from $D$ to a much smaller $\tilde D$.
> \textbf{Our goal is to prune the attention output channels while maintaining and even improving the prediction accuracy of the original transformer.}
> However, directly reducing the channels attention outputs, even by carefully designed methods, would adversely affect the performance of the model.
> In this paper, we propose to maintain or even improve the prediction accuracy of a visual transformer with pruned attention outputs channels
> by computing a more informative affinity matrix $W$ through selecting informative channels in the query $Q$ and the key $V$. That is, only selected columns of $Q$,
> which correspond to the same selected rows of $K^{\top}$, are used to compute the affinity matrix $W  = \sigma(QK^{\top})$, which is refered to as channel selection
> for attention weights and illustrated in Figure 1a. We note that the attention outputs, which are also the input features to the MLP, is $W\times V$,
> and every input feature to the MLP is an aggregation of the rows of the value $V$ using the attention weights in $W$.
> As a result, pruning the channels of $W\times V$ amounts to pruning the channels of $V$ in the weighted aggregation. If the affinity $W$ is more informative,
>  it is expected that a smaller number of features (rows) in $V$ contribute to such weighted aggregation, and the adverse effect of channel selection on the prediction accuracy
>  of the transformer network is limited. Importantly, with a very informative affinity $W$, every input feature of the MLP is obtained by aggregation of the most relevant features (rows)
>  in $V$, which can even boost the performance of visual transformers after channel selection or pruning of the attention outputs.
>
>
> The idea of channel selection for the attention weights can be viewed from the perspective of Information Bottleneck (IB).
> Let $X$ be the input training features, $\tilde X$ be the learned features by the network, and $Y$ be the ground truth training labels for a classification task.
> The principle of IB is maximizing the mutual information between $\tilde X$ and $Y$ while minimizing the mutual information between $\tilde X$ and $X$. That is,
> IB encourages the network to learn features more correlated with the class labels while reducing their correlation with the input. Extensive empirical and theoretical
> works have evidenced that models respecting the IB principle enjoys compelling generalization. With channel selection for the attention weights, every feature in the attention
> outputs aggregates less features of the value $V$, so the attention outputs are less correlated with the training images so the IB principle is better adhered. This is reflected
> in Table 5 in Section C.2 of the supplementary, where a model for ablation study with channel selection for attention weights,
> DCS-Arch1 w/o IB Loss, enjoys less IB loss and higher top-1 accuracy than the vanilla transformer, MobileViT-S. It is noted that the model, DCS-Arch1 w/o IB Loss, only uses
> the regular cross-entropy loss in the retraining step, and smaller IB loss indicates that the IB principle is better respected. In order to further decrease the IB loss,
> we propose an Information Bottleneck (IB) inspired channel selection for the attention weights $\mathbf W$ where the learned attention weights can be more informative by explicitly
> optimizing the IB loss for visual transformers. Our model termed ``DCS-MobileViT-S'' in Table 5 is the visual transformer with the IB loss optimized,
> so that more informative attention weights are learned featuring even smaller IB loss and even higher top-1 accuracy.

---

> > ### Author Response · Authors · 2023-11-23
> > **Response to Reviewer wQJ5 Part 2**
> >
> > **The Regular Cross-Entropy Loss Cannot Decrease the IB Loss Enough**
> >
> > We show the results of the vanilla visual transformers, their ablation study models marked with `(w/o IB Loss)'', and the corresponding DCS-Tranformer models in the table below.
> > The models marked by ``(w/o IB Loss)'' use regular cross-entropy without IB loss.
> > It can be observed that while the regular cross-entroly loss can decrease the IB loss so that the IB principle,
> > learning features more strongly correlated with class labels while decreasing their correlation with the input, is adhered, the IB loss can be decreased further to a considerable extent
> > by optimizing the IB loss explicitly. In particular, our DCS-Transformer models improves the vanilla visual transformers and their ablation study models by a large margin in terms of
> > both IB loss and top-1 accuracy.
> >
> > **Table 1** Parameters, FLOPs, Top-1, IB Loss of vanilla visual transformers, their ablation study models and the corresponding DCS-Tranformer models
> > | Models                                                       | \# Params | FLOPs | Top-1 | IB Loss   |
> > | :----------------------------------------------------------: | :-------: | :---: | :---: | :---------: |
> > | MobileViT-S                                                  | 5.6      | 1.6  | 78.4 | -0.00432 |
> > | DCS-MobileViT-S (w/o IB Loss)                                | 4.8      | 1.2  | 79.9 | -0.00475 |
> > | **DCS-MobileViT-S** | **4.8**  | **1.2** | **81.0** | **-0.05122** |
> > | MobileViT-XS                                                 | 2.3      | 0.71 | 74.8 | -0.00419 |
> > | DCS-MobileViT-XS (w/o IB Loss)                               | 2.0      | 0.54 | 75.7 | -0.00466 |
> > | **DCS-MobileViT-XS**                                         | **2.0**  | **0.54** | **76.8** | **-0.04918** |
> > | EfficientViT (r224)                                          | 9.1      | 0.52 | 79.4 | -0.00451 |
> > | DCS-EfficientViT (r224) (w/o IB Loss)                        | 8.2      | 0.46 | 80.0 | -0.00507 |
> > | **DCS-EfficientViT (r224)**                                  | **8.2**  | **0.46** | **81.2** | **-0.06102** |
> >
> >
> >
> > (2) **Significantly Improved Results of DCS-Transformer**
> >
> > We have improved the results of DCS-Transformer by an improved training strategy which tunes the hyperparameters $\lambda,\eta,t_{\texttt{warm}}$ by cross-validation. The details are presented in
> > Section B of the supplementary of the revised paper. The value of $\lambda$ is selected from $\{0.1, 0.15, 0.2, 0.25, 0.3, 0.35, 0.4, 0.45, 0,5\}$.
> > The value of $\eta$ is selected from $\{0.1, 0.5, 1, 5, 10, 50, 100\}$. The value of $t_{\text{warm}}$, which is the number of warm-up epochs, is selected from $\{0.1t_{\texttt{train}}, 0.2t_{\texttt{train}}, 0.3t_{\texttt{train}}, 0.4t_{\texttt{train}}, 0.5t_{\texttt{train}}, 0.6t_{\texttt{train}}\}$,
> > where $t_{\texttt{train}}=300$ is the total number of training epochs.
> > We select the values of $\eta$, $\lambda$, and $t_{\texttt{warm}}$ that
> > lead to the smallest validation loss. It is revealed that $t_{\texttt{warm}} = 90$ is chosen for all the three visual transformers in our experiments.
> > Using the searched hyperparameters by cross-validation which is a principled method for tuning hyperparameters, we have obtained significantly improved results of
> > DCS-Transformers shown in Table 1, and the same results are also reflected in the revised paper. For exmaple, compared to the top-1 accuracy of
> > 79.9% of DCS-MobileViT-S (w/o IB Loss), DCS-MobileViT-S achieves a top-1 accuracy of 81.0%.
> >
> > (3) **Improved Readability of the Code**
> >
> > We have revised our code and $\eta$ is now initialzied to $50$ in the code:
> > group.add_argument('--ib-weight', type=float, default=50, help='weight of IB loss (default: 50)').
> > We would like to emphasize that $\eta = 50$ is for the DCS-Transfformers. All the other baseline models do not use IB loss
> > (so $\eta = 0$  equivalently). Again,
> > all the hyperparameters are searched by cross-validation and the best hyperparameters with smallest validation loss are chosen,
> > and this process is detailed in Section B of the supplementary.
> >
> >
> > (4) **Discussion on Related Works about Neural Architecture Search (NAS)**
> >
> > We have added discussion on related works about NAS in Section 3.4 of the revised paper with the mentioned seminal works cited. Thank you for your suggestion!

---

### Official Review · Reviewer_Q1sT · 2023-10-31

**Soundness:** 3 good
**Presentation:** 3 good
**Contribution:** 3 good
**Rating:** 6
**Confidence:** 5

**Summary:**

This paper introduces a compact transformer architecture exploring the differentiable channel selection. There are two types of channel selection, which are channel selection for attention weights and channel selection for attention outputs. In addition,  the IB loss is employed to boost the performance of the proposed framework. Extensive experiments on image classification and object detection verifies the effectiveness of the proposed method.

**Strengths:**

1. The paper is well-written and well-motivated.

2. The design of the channel selection for attention weights and attention outputs make sense, corresponding to the matrix multiplication for attention and MLP. The IB loss is further considered to improve the performance.

3. The comparison with the SOTA pruning methods and compact models show that the proposed method is effective on the mobile devices.

**Weaknesses:**

1. There are only comparisons of parameters and FLOPs, I wonder the actual inference time of the proposed method.

2. In Figure 2, there are two points for EfficientViT while only one point for DCS-EfficientViT. What's the performance of another point? In another word, is the proposed method still valuable for a larger model?

3. The hyper-parameters are carefully designed such as the temperature etc. I am doubt about the generalization of the proposed method.

**Questions:**

See weaknesses.

---

> ### Author Response · Authors · 2023-11-23
> **Response to Reviewer Q1sT**
>
> We appreciate the review and the suggestions in this review. The raised issues are addressed below.
>
> (1) **Actual Inference Time**
>
> We compare DCS-Transformer to the current state-of-the-art pruning methods for visual transformers, the results are shown in the table below.
> DCS-Transformer is compared to S$^2$ViTE[2], SPViT[3] and SAViT [4] on EfficientViT-B1 (r224) [1].
> For S$^2$ViTE[2], SPViT[3] and SAViT [4], the pruning is performed on the ImageNet training data. After obtaining the pruned networks, we fine-tune the pruned networks using the same setting as [1].
> It can be observed that DCS-EfficientViT-B1 outperforms all pruned models by at least 1.9% in top-1 accuracy with even less parameters and FLOPs.
> We report the actual inference time of all the model on a NVIDIA V100 GPU.
>
>
> | Methods                          | \# Params | FLOPs   | Inference Time (ms/image) | Top-1 |
> | :------------------------------: | :-------: | :-----: | :---: | :---: |
> | EfficientViT-B1 (r224) [1]       | 9.1 M    | 0.52 G | 2.654 | 79.4 |
> |S$^2$ViTE-EfficientViT-B1 (r224) [2] | 8.2 M    | 0.47 G | 2.438 | 79.0 |
> | SPViT-EfficientViT-B1 (r224) [3]  | 9.2 M    | 0.49 G | 2.451 | 79.3 |
> | SAViT-EfficientViT-B1 (r224) [4] | 8.4 M    | 0.47 G | 2.435 | 79.2 |
> | DCS-EfficientViT-B1 (r224)       | 8.2 M    | 0.46 G | 2.427 | 81.2 |
>
>
>
> (2) **Completed Figure 2**
>
> The two points for DCS-EfficientViT have been added to Figure 2 in the revised paper.
>
> (3) **Tuning Hyperparameters by Cross-Validation**
>
> Thank you for your concern about the hyperparameters. We set the initial value of the temperature $\tau$ to $4.5$ and decrease it by a factor of $0.95$ every epoch, and this is the setting used for all the experiments in this paper. In the revised paper, DCS-Transformer is trained by an improved training strategy
> which tunes the hyperparameters $\lambda,\eta,t_{\texttt{warm}}$ by cross-validation. The details are presented in
> Section B of the supplementary of the revised paper. The value of $\lambda$ is selected from $\{0.1, 0.15, 0.2, 0.25, 0.3, 0.35, 0.4, 0.45, 0,5\}$.
> The value of $\eta$ is selected from $\{0.1, 0.5, 1, 5, 10, 50, 100\}$. The value of $t_{\text{warm}}$, which is the number of warm-up epochs, is selected from $\{0.1t_{\texttt{train}}, 0.2t_{\texttt{train}}, 0.3t_{\texttt{train}}, 0.4t_{\texttt{train}}, 0.5t_{\texttt{train}}, 0.6t_{\texttt{train}}\}$,
> where $t_{\texttt{train}}=300$ is the total number of training epochs.  We select the values of $\eta$, $\lambda$, and $t_{\texttt{warm}}$ that
> lead to the smallest validation loss. It is revealed that $t_{\texttt{warm}} = 90$ is chosen for all the three visual transformers in our experiments.
> Using the searched hyperparameters by cross-validation which is a principled method for tuning hyperparameters, we have obtained significantly improved results of
> DCS-Transformers shown in Table 1, and the same results are also reflected in the revised paper. For exmaple, compared to the top-1 accuracy of
> 79.9% of DCS-MobileViT-S (w/o IB Loss), DCS-MobileViT-S achieves a top-1 accuracy of 81.0%.
>
> **References**
>
> [1] Cai et al. "Efficientvit: Enhanced linear attention for high-resolution low-computation visual recognition." ICCV 2023.
>
>
> [2] Chen, Tianlong, et al. "Chasing sparsity in vision transformers: An end-to-end exploration." NeurIPS 2021.
>
>
> [3] Kong, Zhenglun, et al. "SPViT: Enabling faster vision transformers via soft token pruning." ECCV 2022.
>
>
> [4] Zheng, Chuanyang, et al. "SAViT: Structure-Aware Vision Transformer Pruning via Collaborative Optimization." NeurIPS 2022.

---

### Official Review · Reviewer_JKXw · 2023-11-06

**Soundness:** 3 good
**Presentation:** 2 fair
**Contribution:** 3 good
**Rating:** 6
**Confidence:** 3

**Summary:**

This paper proposes a DCS mechanism, which achieves network pruning via differentiable channel selection. Specifically, two channel selection strategies have been proposed, that is, channel selection for attention weights and channel selection for attention outputs. To ensure that only informative channels have been selected, the authors incorporate IB loss, which is inspired by the information bottleneck theory. Experiments on image classification and object detection have demonstrated the effectiveness of the proposed method, as well as its generalization on multiple Transformer architectures, including EfficientViT and MobileViT.

**Strengths:**

Basically, the main contribution of the proposed method is two-fold: a straightforward channel selection mechanism, and the intuitive incorporation of information bottleneck theory. Although both ideas have long been proposed, their combination and utilization in Transformer pruning may still be inspiring, especially to researchers in this specific field. The authors' claims have also been well-supported by the extensive experimental results. The manuscript is generally well-written and the English usage is satisfactory.

**Weaknesses:**

There are several aspects of this work that could be further improved:

1. The authors may consider focusing more on illustrating their motivation and ideas instead of explaining the technical details. For example, the usage of information bottleneck in the proposed method is not well-introduced. I was expecting to see how the information bottleneck theory is integrated into the proposed architecture and the rationale behind it, but only to fine detailed derivation of the variational upper bound for the IB loss.

2. Is it possible that the propose module be applied to more classical architecture of ViT, e.g., the vanilla ViT or Swin? And what would be performance if DCS is used in semantic segmentation tasks. More experimental results would make the paper more convincing.

3. It seems that the authors fail to compare their method to other pruning techniques, but only show that DCS is effective as it successfully reduce the number of parameters without sacrificing the performance. I also expect a comprehensive comparison against benchmark pruning methods in term of the overall computational cost.

After rebuttal:
I appreciate the detailed response provided by the authors, which have solved much of my concern and lead to the increase of overall rating. I would recommend the authors to integrate the supplementary results they provide in the rebuttal phase into their manuscript, so as to make it more intuitive and convincing.

**Questions:**

Please refer to the weakness. My concern mainly lies in the experiment section.

---

> ### Author Response · Authors · 2023-11-23
> **Response to Reviewer JKXw Part 1**
>
> We appreciate the review and the suggestions in this review. The raised issues are addressed below.
>
> (1) **More Detailed Motivation**
>
> More detailed motivation and ideas for the incorporate of information bottleneck in the proposed architecture and
> the rationale behind it are presented in Section 1 of the revised paper. It is attached below for your convenience.
>
> **Motivation**
>
> A typical transformer block can be written as
> $\texttt{Output} = \texttt{MLP} \left(\sigma(QK^{\top})\times V \right)$
> where $Q,K,V \in \mathbb{R}^{N \times D}$ denote the query, key, and value respectively
> with $N$ being the number of tokens and $D$ being the input channel number.
> $\sigma(\cdot)$ is an operator, such as Softmax, which generates the attention weights or affinity between the tokens.
> We refer to $W  = \sigma(QK^{\top}) \in \mathbb{R}^{N \times N}$ as the affinity matrix between the tokens.
> $\textup{MLP}$ (multi-layer perceptron network) generates the output features of the transformer block.
> There are $D$ channels in the input and output features of the MLP, and $D$ is also the channel of the attention outputs.
> Due to the fact that the MLP accounts for a considerable amount of FLOPs in a transformer block, the size and FLOPs of a transformer
> block can be significantly reduced by reducing the channels of the attention outputs from $D$ to a much smaller $\tilde D$.
> \textbf{Our goal is to prune the attention output channels while maintaining and even improving the prediction accuracy of the original transformer.}
> However, directly reducing the channels attention outputs, even by carefully designed methods, would adversely affect the performance of the model.
> In this paper, we propose to maintain or even improve the prediction accuracy of a visual transformer with pruned attention outputs channels
> by computing a more informative affinity matrix $W$ through selecting informative channels in the query $Q$ and the key $V$. That is, only selected columns of $Q$,
> which correspond to the same selected rows of $K^{\top}$, are used to compute the affinity matrix $W  = \sigma(QK^{\top})$, which is refered to as channel selection
> for attention weights and illustrated in Figure 1a. We note that the attention outputs, which are also the input features to the MLP, is $W\times V$,
> and every input feature to the MLP is an aggregation of the rows of the value $V$ using the attention weights in $W$.
> As a result, pruning the channels of $W\times V$ amounts to pruning the channels of $V$ in the weighted aggregation. If the affinity $W$ is more informative,
>  it is expected that a smaller number of features (rows) in $V$ contribute to such weighted aggregation, and the adverse effect of channel selection on the prediction accuracy
>  of the transformer network is limited. Importantly, with a very informative affinity $W$, every input feature of the MLP is obtained by aggregation of the most relevant features (rows)
>  in $V$, which can even boost the performance of visual transformers after channel selection or pruning of the attention outputs.
>
> The idea of channel selection for the attention weights can be viewed from the perspective of Information Bottleneck (IB).
> Let $X$ be the input training features, $\tilde X$ be the learned features by the network, and $Y$ be the ground truth training labels for a classification task.
> The principle of IB is maximizing the mutual information between $\tilde X$ and $Y$ while minimizing the mutual information between $\tilde X$ and $X$. That is,
> IB encourages the network to learn features more correlated with the class labels while reducing their correlation with the input. Extensive empirical and theoretical
> works have evidenced that models respecting the IB principle enjoys compelling generalization. With channel selection for the attention weights, every feature in the attention
> outputs aggregates less features of the value $V$, so the attention outputs are less correlated with the training images so the IB principle is better adhered. This is reflected
> in Table 5 in Section C.2 of the supplementary, where a model for ablation study with channel selection for attention weights,
> DCS-Arch1 w/o IB Loss, enjoys less IB loss and higher top-1 accuracy than the vanilla transformer, MobileViT-S. It is noted that the model, DCS-Arch1 w/o IB Loss, only uses
> the regular cross-entropy loss in the retraining step, and smaller IB loss indicates that the IB principle is better respected. In order to further decrease the IB loss,
> we propose an Information Bottleneck (IB) inspired channel selection for the attention weights $\mathbf W$ where the learned attention weights can be more informative by explicitly
> optimizing the IB loss for visual transformers. Our model termed ``DCS-MobileViT-S'' in Table 5 is the visual transformer with the IB loss optimized,
> so that more informative attention weights are learned featuring even smaller IB loss and even higher top-1 accuracy.

---

> ### Author Response · Authors · 2023-11-23
> **Response to Reviewer JKXw Part 2**
>
> (2) **DCS-Transformer for ViT and Swin, and DCS-Transformer for the Task of Segmentation**
>
> The proposed DCS-Transformer module can be straightforwardly applied to a broader class of visual transformers beyond the examples covered in this paper, including
> the vanilla ViT [1] and Swin [2]. This is because the two components of a DCS-Transformer, channel selection for attention weights and
> channel selection for attention outputs can be directly applied to ViT and Swin. Here we replace all the transformer blocks in ViT-S/16 [1] and Swin-T [2], obtaining DCS-ViT-S/16 and DCS-Swin-T respectively.
> We use the same settings for search and training as described in Section 4.1 of our paper.
> All the models are trained on the training set of ImageNet-1K and tested on the validation set of ImageNet-1K. The results are shown in the Table below.
>
>
> |    Models    | # Params | FLOPs | Top-1 |
> | :----------: | :------: | :---: | :---: |
> | ViT-S/16 [1] |  22.1 M  | 9.2 G | 81.2  |
> | DCS-ViT-S/16 |  20.2 M  | 8.3 G | **82.8**  |
> |  Swin-T [2]  |  29.0 M  | 4.5 G | 80.6  |
> |  DCS-Swin-T  |  26.1 M  | 4.0 G | **82.1**  |
>
>
> In Section E of the supplementary of the revised paper, we present the results of DCS-EfficientViT-B1 for the task of instance segmentation on the COCO dataset
> under the same setting as [3]. Please refer to Section E for more details. The results in Table 7 of the revised paper are copied to the table below for your convenience.
> We report the mean bounding box Average Precision (APb) and mean mask Average Precision (APm) as well as bounding box Average Precision (APb)
> and mask Average Precision (APm) under IoU thresholds of 0.5 and 0.75. It can be observed that DCS-EfficientViT-B1 consistently improves the performance of segmentation across various thresholds.
>
>
> | Methods           | $AP^{box}$ | $AP^{box}_{50}$ | $AP^{box}_{75}$ | $AP^{mask}$ | $AP^{mask}_{50}$ | $AP^{mask}_{75}$ |
> | :---------------: | :----: | :----: | :----: | :----: |:----:|:----:|
> | EViT[3]               | 32.8   | 54.4 | 34.5 | 31.0    | 51.2 | 32.2 |
> | EfficientViT-B1 [4]     | 33.5   | 55.4 | 34.8 | 31.9    | 52.3 | 32.7 |
> | DCS-EfficientViT-B1 | 34.8   | 56.3 | 35.3 | 33.2    | 53.1 | 33.3 |
>
>
>
>
>
>
> (3) **Comparison with Pruning Methods**
>
> We compare DCS-Transformer to the current state-of-the-art pruning methods for visual transformers, the results are shown in the table below. DCS-Transformer is compared to S$^2$ViTE[2], SPViT[3], and SAViT [4] on EfficientViT-B1 (r224) [1].
> For S$^2$ViTE[2], SPViT[3] and SAViT [4], the pruning is performed on the ImageNet training data. After obtaining the pruned networks, we fine-tune the pruned networks using the same setting as [1].
> It can be observed that DCS-EfficientViT-B1 outperforms all pruned models by at least 1.9% in top-1 accuracy with even less parameters and FLOPs.
>
>
> We also include actual inference time of all the model on a NVIDIA V100 GPU.
>
>
> | Methods                          | \# Params | FLOPs   | Inference Time (ms/image) | Top-1 |
> | :------------------------------: | :-------: | :-----: | :---: | :---: |
> | EfficientViT-B1 (r224) [4]       | 9.1 M    | 0.52 G | 2.654 | 79.4 |
> |S$^2$ViTE-EfficientViT-B1 (r224) [5] | 8.2 M    | 0.47 G | 2.438 | 79.0 |
> | SPViT-EfficientViT-B1 (r224) [6]  | 9.2 M    | 0.49 G | 2.451 | 79.3 |
> | SAViT-EfficientViT-B1 (r224) [7] | 8.4 M    | 0.47 G | 2.435 | 79.2 |
> | DCS-EfficientViT-B1 (r224)       | 8.2 M    | 0.46 G | 2.427 | 81.2 |
>
>
> **References**
>
> [1] Dosovitskiy et al. "An image is worth 16x16 words: Transformers for image recognition at scale." ICLR 2021.
>
> [2] Liu et al. "Swin transformer: Hierarchical vision transformer using shifted windows." ICCV, 2021.
>
> [3] Liu et al. "EfficientViT: Memory Efficient Vision Transformer with Cascaded Group Attention." CVPR 2023.
>
> [4] Cai et al. "Efficientvit: Enhanced linear attention for high-resolution low-computation visual recognition." ICCV 2023.
>
>
> [5] Chen, Tianlong, et al. "Chasing sparsity in vision transformers: An end-to-end exploration." NeurIPS 2021.
>
> [6] Kong, Zhenglun, et al. "SPViT: Enabling faster vision transformers via soft token pruning." ECCV 2022.
>
> [7] Zheng, Chuanyang, et al. "SAViT: Structure-Aware Vision Transformer Pruning via Collaborative Optimization." NeurIPS 2022.

---

### Meta-Review · Area_Chair_pqrH · 2023-12-05

**Metareview:**

This submission received two positive scores and one negative score. After reading the paper, the review comments and the rebuttal, the AC thinks the major concerns about the motivation and contribution remain. After carefully reading the paper, the review comments, the AC deemed that the paper should undergo a major revision, thus is not ready for publication in the current form.

**Justification For Why Not Higher Score:**

N/A

**Justification For Why Not Lower Score:**

N/A

---

### Decision · Program_Chairs · 2024-01-16

Reject